# Post-pleistocene colonisation rather than the contemporary environment has most influenced the current population structure of Scottish Atlantic salmon (*Salmo salar*)

Finn Cowell[1,2]*, Oscar E. Gaggiotti[1], Eef Cauwelier[3]

1 Centre for Biological Diversity, School of Biology, University of St Andrews, St Andrews, Fife, Scotland, United Kingdom, 2 Institute for Biodiversity and Freshwater Conservation, University of the Highlands and Islands, Inverness, Scotland, United Kingdom, 3 Marine Directorate, Freshwater Fisheries Laboratory, Faskally, Pitlochry, Perthshire, Scotland, United Kingdom

* EX01FC@uhi.ac.uk

## Abstract

Genetic structuring in populations is the result of both historical and contemporary environmental factors driving genetic drift, natural selection and gene flow, as well as purely genetic factors, such as mutation and recombination. In Atlantic salmon (*Salmo salar*), re-colonisation of rivers after the last Ice Age was shown to be an important factor in shaping contemporary population structure, though the observed structure was more complex than was predicted through founder effects. Thus, other, perhaps more contemporary factors may also play a role. Here, we investigated the influence of the time since deglaciation, distance to the sea, population connectivity, temperature, water quality, waterbody modifications, and environmental protections on spatial structuring of genetic diversity, based on microsatellite data (33 loci) collected from 48 Scottish *S. salar* populations. The results confirmed that recently deglaciated areas are less genetically diverse and more differentiated. Modified waterbodies also exhibit less genetic diversity and greater differentiation, although this effect differs between rivers draining on the east and west coasts of Scotland. Distance to the sea also had a non-negligible effect, while the other considered factors did not have a significant effect.

## Introduction

Genetic relationships between populations are shaped by processes that either increase differentiation, such as natural selection, genetic drift, founder effects and mutation or decrease it, such as gene flow through migration. Understanding how these processes have influenced contemporary populations is of great interest to those involved in conservation biology as intraspecific diversity is an important component of biodiversity, especially at high latitudes [1].

**Data availability statement:** All relevant data are within the manuscript and its Supporting information files.

**Funding:** The author(s) received no specific funding for this work.

**Competing interests:** The authors have declared that no competing interests exist.

At high latitudes, palearctic fishes, including anadromous Salmonids, have colonised and re-colonised rivers following the respective retreat and expansion of ice sheets [2]. Indeed, the importance of ice sheets on anadromous Salmonid populations has been reported in the Baltic Sea [3], Iceland [4] the British Isles [5,6], Norway and the areas of Russia surrounding the Barents and White seas [7]. This observed population genetic structure has been attributed to strong founder effects upon colonisation of newly ice-free rivers [8]. Furthermore, anadromous Salmonids, such as the Atlantic salmon (*Salmo salar*) display high levels of philopatry which may result in relatively low levels of gene flow, allowing mutations to accumulate, genetic drift to randomly fix alleles and natural selection to lead to adaptation to the unique local environment [9–11].

Initial colonisation and associated establishment of populations will have been further influenced by factors related to deglaciation, such as isostatic rebound, which would have affected river accessibility [12,13] and population connectivity. Isostatic rebound resulted in relative sea levels decreasing [13] resulting in some of the lower stretches of contemporary rivers being saltwater areas that were not colonisable by *S. salar* until long after the ice sheets melted. Concurrently, higher reaches of rivers, with lower carrying capacity, might have become available as the ice sheet retreated, expanding the species range upstream. It is therefore uncertain how the gradual expansion within a river affected the relationship between distance to sea and differentiation, as well as population connectivity.

Despite Cauwelier *et al.* [5] identifying the melting of ice sheets as having an important influence on genetic structuring on Scottish *S. salar* populations, ecological niche modelling could not explain all the differences between populations. Contemporary factors, such as climate change, waterbody modification, etc. are, therefore, of interest, as there have been many anthropogenic impacts on watercourses since the time of deglaciation and it is possible that these have resulted in genetic changes in *S. salar* populations.

The depredation of fishes by human beings is well known to have begun in prehistory but this has intensified during the late modern period [14]. River temperatures documented in Scotland during the 20th and 21st century have risen, likely a result of anthropogenic climate change [15]. The water quality itself has been impacted greatly as well and the riverscape has been changed by humans to allow for hydroelectricity generation, irrigation of farmland and the prevention of flooding [16]. All these factors have prompted environmental designations to be put in place in order to protect rivers and the organisms that live there.

There exist a number of reasons why one might expect the structure of *S. salar* to be impacted by human activities. In some cases, local adaptation to thermal regimes has been shown to result in greater genetic diversity within and differentiation between *S. salar* populations [17]. However, the reported effect of temperature may be indirect rather than causal, as it affects other elements of the ecosystem, such as bacterial diversity [18]. Poor water quality has been shown to restrict distribution [19] and reduce abundance [20] in anadromous fishes, the former having the potential to reduce gene flow, while the latter could increase genetic drift. Further direct and

indirect effects of anthropogenic activity, such as modifications to waterways, e.g., dams and water abstraction have been shown to reduce genetic diversity and increase genetic differentiation [21–23]. Meanwhile, population interbreeding due to stocking and/or farm escapees associated with aquaculture also impact genetic structuring leading to less genetic differentiation between populations [24–27].

Lastly, efforts to protect *S. salar* populations are expected to also have an impact on diversity and structuring, whether through stocking and/or habitat conservation/restoration. In Scotland many rivers are designated as SACs (Special Area of Conservation) with this designation often being given specifically to protect *S. salar* habitat. The impact of these designations on *S. salar* populations is, however, poorly understood. One might expect an increase in population size in response to these measures which would, in turn, result in reduced levels of genetic drift.

All the above factors, both historical and contemporary, have the potential to alter levels of gene flow and genetic drift. These evolutionary forces can, in turn, influence the genetic composition of extant populations. However, the possible relationship between genetic differentiation and environmental factors, both historical and contemporary, has not been considered concurrently. Here, we investigated the effect of both historical and contemporary factors on Scottish *S. salar* population structure. Data for 48 *S. salar* populations from across Scotland were used to calculate measures of genetic differentiation and diversity. The impacts of the time since deglaciation, distance to the sea, population connectivity, temperature, water quality, waterbody modification, and environmental protections were assessed through general linear modelling using both a frequentist and a Bayesian approach.

## Methods

### Sample collection and screening

Due to computational demands required for the Bayesian approach, the full dataset described in Cauwelier *et al.* [5] was not used. 48 sites were randomly selected to represent all genetic groups identified by STRUCTURE analysis in Cauwelier *et al.* [5], with numbers proportional to the geographic area of the group (Fig 1). This dataset was extended by screening for an additional sixteen microsatellites, in order to increase genome coverage and potentially enhance resolution.

Between 2003 and 2010, 48 sites across 41 Scottish rivers were electrofished for juvenile *S. salar* (Fig 1). The care and use of experimental animals complied with Scottish Government animal welfare laws, guidelines and policies as approved by Marine Scotland Science AWERB under Home Office project licence PP3525229. Exact field methodology may have varied slightly from river to river but, in brief, fish anaesthetics, such as tricaine methanesulphonate (MS-222), were mixed with a set volume of water in a bucket dedicated for this purpose. Fin clips were taken from these anaesthetised fish and were then allowed to recover in a bucket of aerated water, before being released and clips stored in 99% ethanol. DNA was extracted using the method described by Knox *et al.* [28] and initially amplified for three multiplexes encompassing fourteen microsatellites ([29]- excluding *SsaD486*) plus *SsaD48*, *SsaD71* [30] and *SP1608* [31]. Full details on PCR conditions and fragment analysis are described in Cauwelier *et al.* [8]. Screening for an additional sixteen microsatellites (*EST107*, *EST19*, *EST28*, *EST68*, *MHC1*, *MHC2*, *Sleel53*, *Sleen82*, *Ssa405*, *Ssa407*, *Ssa412*, *Ssa98*, *Ssleer15*, *SsOSL25*, *Ssosl85* and *SsSP2215*) was carried out at the Norwegian Institute for Marine Research, using the protocol described in Harvey *et al.* [32]. The resulting dataset consisted of 1,044 fish genotyped at 33 microsatellite loci.

### Genetic analysis

Tests for deviation from Hardy-Weinberg (HW) equilibrium were conducted using the Markov chain method in the Genepop package [34] in R version 4.0.3 [35]. This was run with 2,000 iterations of 500 batches with a dememorization length of 10,000. The same package was used to test for linkage disequilibrium (LD), with 1,000 iterations over 100 batches and a dememorization of 10,000. Handling multiple tests to avoid false positives was done by following Waples [36], using the cumulative binomial distribution to identify if the number of positive tests significantly exceeded those expected due to chance alone.

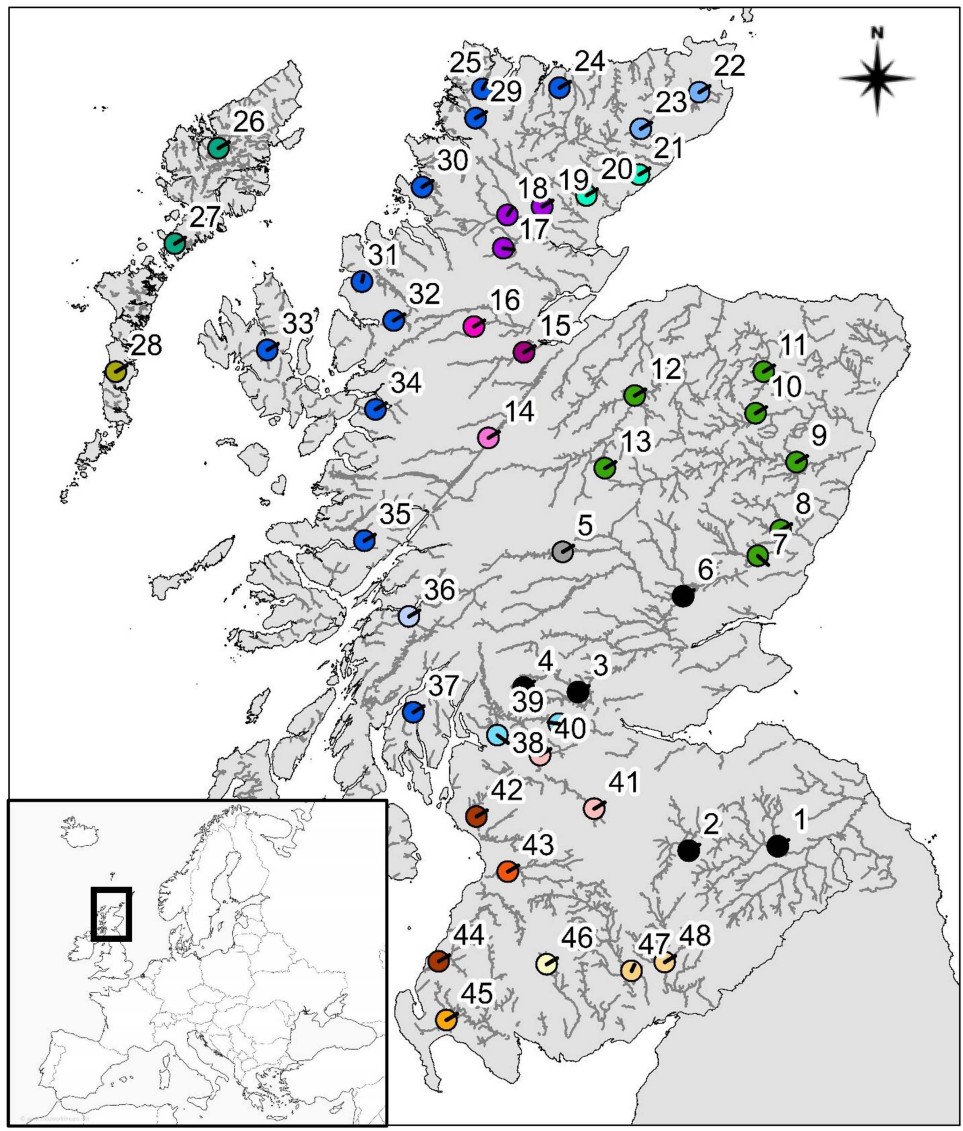

**Fig 1. Map of Scotland (1: 2,500,000) and its location within Europe (inset), depicting the electrofished sites [33].** Numbers relate to site numbers detailed in Table 1 and the colours refer to the genetic cluster to which each site belongs, as found by Cauwelier *et al.* [5]. Source of map of Europe: www.freeworldmaps.net, map of Scotland: https://osdatahub.os.uk/downloads/open/BoundaryLine and salmon rivers: https://marinescotland.atkinsgeo-spatial.com/nmpi/default.aspx?layers=843.

Rarefied allelic richness was calculated using hierfstat [37]. Population-specific differentiation was based on $F_{st}$ values, which were calculated using GESTE [38]. GESTE is a Bayesian programme that estimates population-specific $F_{st}$ values measuring genetic divergence between each population and the metapopulation as a whole.

### Environmental data

Environmental data were synthesised from several sources. Data on water quality and protection status were obtained from NatureScot and the Scottish Environmental Protection Agency (SEPA) [39]. Data on the water quality in rivers used the Water Framework Directive (WFD) classification of a waterbody as either High, Good, Moderate, Poor or Bad, which

was based on many factors relating to chemistry, biology, specific pollutants and hydromorphology. The WFD also details the extent to which a river has been modified by human activities, a *heavily modified* river being one which has been permanently physically altered in a way which has substantially changed its character, and as such, is designated under article 4(3) of the WFD. NatureScot provided information on the protection status of rivers and surrounding areas. The designated protection for each site was recorded and, if the reason for this designation included the need to protect the spawning and nursery habitat of *S. salar*, then this information was also recorded. The qualitative descriptors of habitat protection were coded as three binary dummy variables. The first variable showed if *S. salar* were specially protected or not in that section of river, the second detailed if the section of river was protected for a reason unrelated to *S. salar* or not, and the third detailed if the river, as well as a broader area outwith the riparian zone, was protected or not.

Water temperature data came from the Scotland River Temperature Monitoring Network, through the Marine Scotland website [40]. These are modelled data that have been inferred from strategic monitoring of temperature at specific sites throughout Scotland and the consideration of influential landscape features [41]. The figures used were the predictions of the maximum daily river temperature for the hottest year in the last 20 years.

The time since glaciation was also considered, using maps of the deglaciation process made by Cauwelier *et al.* [5]. Distance to the sea and elevation data (OS terrain 50 layer) were calculated using ArcMap 10.6 [42]. Pairwise distances were calculated by measuring the shortest swimmable distance from the mouth of the rivers using Free Map Tools [43]. This was then added to the distance to the sea measures of the two rivers. This pairwise distance matrix was then converted into a population-specific measure of connectivity, which was calculated as the mean of all pairwise distances between a population and all other populations.

In order to limit issues arising from multicollinearity, a correlation matrix was calculated between all the environmental variables. One variable of each pair was removed in cases of a correlation coefficient greater than 0.8. Furthermore, variables with a variance inflation factor (VIF) greater than 5 were also removed. Distance to the sea was used in place of elevation for this reason.

## Modelling

Rarefied allelic richness was analysed by using General Linear Models (GLMs), with all environmental variables initially included. A procedure of backwards elimination was employed, using a p-to-remove criterion, until only a single independent variable was included in the remaining model. The best model was then chosen as the one with the lowest corrected Akaike's Information Criterion (AICc) score. All GLMs assumed a normal distribution, as this best fitted the model assumptions. As many of the variables were categorical, the *drop1* function in R was used, as this added and removed each term of the model in turn, allowing the effect of that factor on the $R^2$ to be realised and the significance to be determined.

GESTE was used to investigate the association between population-specific $F_{st}$ and environmental factors following the procedure described by Gaggiotti *et al.* [44]. This involved using default settings except for using ten pilot runs of 1,000 iterations, with an additional burn in period of 5,000,000 iterations. A thinning interval of 50 was then used during the main run. An initial exploratory run with all factors (full model) was used to identify the five factors with the highest marginal posterior probability across all models, which were then used in a second run. The five models with the highest posterior probabilities were reported.

For comparison with the GESTE connectivity effect, a pairwise $F_{st}$ matrix was generated using Genalex 6.5 [45,46]. In an effort to detect any pattern of isolation-by-distance, this matrix was linearised ($F_{st}/(1-F_{st})$), as suggested by Rousset [47] and was regressed against a geographical distance matrix with a classic mantel test using the R package ade4 [48] with 9999 repeats.

Two different spatial scales were analysed for each modelling method. Initially, all rivers were included, which encompassed seven genetic regions [8]. A hierarchical structure was then considered, whereby data were split into rivers draining along the east coast and west coast and analysed separately. Carrying out separate analyses for the two geographic

regions was done in order to avoid confounding effects by factors that could not be accounted for by statistical analyses, given that the anthropogenic effects were likely to differ between the two coasts. In particular, protection of waterbody for a reason other than *S. salar*, as well as protection of the surrounding land, only applied to the west coast. Moreover, there are geomorphological differences between the east and west coasts which have long been recognised [49]. The three sites on the north coast of Scotland were categorised as being in the west coast group, as they form part of a genetic grouping extending west [5].

## Results

### Genetic diversity and differentiation

As Hardy-Weinberg equilibrium deviation and positive tests for LD were not significantly greater than the number expected due to chance alone, all 33 microsatellite loci were analysed. Mean rarefied allelic richness across the 48 Scottish *S. salar* populations equalled 5.42 (SD = 0.26). Population-specific $F_{st}$ generated from GESTE, when all sites were considered, ranged from 0.002 in the River Don (population 10 in Table 1 and Fig 1) to 0.084 in the River Leven (population 39) with a mean $F_{st}$ of 0.021 (SD = 0.019). Full diversity and differentiation results can be found in Table 1.

### General linear modelling

The difference between the AICc score of the simplest model and the next lowest score was not > 2 but it was nonetheless chosen as the best model for its simplicity. Variation in rarefied allelic richness at all sites was best explained by a model considering time since deglaciation (Fig 2 and Table 2). In the case of the east coast, only waterbody modification was included in the best model. Genetic diversity on the west coast meanwhile was best predicted by the sole factor of time since deglaciation. The adjusted $R^2$ for the east coast model was more than twice as high when compared to the all-sites or west coast models.

### GESTE

When predicting genetic differentiation in a model which included all sites, the marginal posterior probabilities of each factor, when GESTE considered all nine (Table 3), was greatest for time since deglaciation. This was also true when considering the east and west coasts separately. However, additionally, on the east coast, waterbody modification and distance to the sea also had relatively high probabilities, and the former had a higher posterior probability than the latter. In all cases, connectivity had a negligible effect; a result corroborated by the classic mantel test between all sites (r = 0.084, p = 0.084).

Subsequent analysis of the effect of the five best environmental factors on population-specific $F_{st}$ across all sites also revealed the predominant influence of time since deglaciation (Table 4) but also found that distance to the sea was an influential factor. Population-specific $F_{st}$ was found to be higher in the more recently deglaciated rivers and lower in sites more distant from the sea (Fig 3). This was also the case when the east coast was considered separately where, additionally, waterbody modification was important (Fig 3, Table 4).

Indeed, unmodified waterbodies exhibited much lower genetic differentiation than those that have been modified. On the other hand, the results from the west coast dataset showed that the null model was best. The marginal posterior probability on the west coast for waterbody modification, distance to the sea and time since deglaciation were 0.099, 0.1 and 0.598, respectively. In this same order, the east coast results were 0.172, 0.152 and 0.567.

## Discussion

The aim of this study was to explore the major historical and contemporary factors contributing to genetic differentiation and diversity in *S. salar*. Overall, the largest factor associated with the observed diversity/differentiation was timing since

 

**Table 1.** Full details of the environmental variables at each site as well as the genetic diversity and differentiation measures.

| Site | Site Name | Drainage System | Rarefied $A_r$ | $F_{st}$ | Time Since Deglaciation | Water Quality | Modified | Salmon Protection | Waterway Protection | Surroundings Protection | Distance to the Sea | Altitude | Connectivity | Temperature | Coast |
|---|---|---|---|---|---|---|---|---|---|---|---|---|---|---|---|
| 1 | River Tweed | River Tweed | 5.67 | 0.009 | −15000 | Good | No | 1 | 0 | 0 | 62 | 95.1 | 624 | 22 | East |
| 2 | Manor Water | River Tweed | 5.7 | 0.007 | −15000 | Good | No | 0 | 0 | 0 | 118 | 254 | 679 | 24 | East |
| 3 | River Teith | River Forth | 5.48 | 0.007 | −15000 | Moderate | No | 1 | 0 | 0 | 7.06 | 13 | 610 | 22 | East |
| 4 | Avon Dhu | River Forth | 5.45 | 0.019 | −12000 | Good | No | 0 | 0 | 0 | 57.4 | 23.6 | 659 | 22 | East |
| 5 | River Tummel | River Tay | 4.87 | 0.061 | −12000 | Good | Yes | 1 | 0 | 0 | 87.4 | 208 | 633 | 21 | East |
| 6 | Lunan Burn | River Tay | 5.57 | 0.015 | −15000 | Moderate | No | 1 | 0 | 0 | 21.8 | 36.1 | 568 | 21 | East |
| 7 | River South Esk | River South Esk | 5.45 | 0.01 | −16000 | Good | No | 1 | 0 | 0 | 22.4 | 54 | 508 | 21 | East |
| 8 | West Water | River North Esk | 5.73 | 0.009 | −16000 | High | No | 0 | 0 | 0 | 17.6 | 50.7 | 494 | 20 | East |
| 9 | Burn of Canny | River Dee | 5.67 | 0.004 | −19000 | Poor | No | 1 | 0 | 0 | 32.7 | 64.2 | 473 | 22 | East |
| 10 | River Don | River Don | 5.69 | 0.002 | −19000 | Good | No | 0 | 0 | 0 | 77.3 | 188 | 513 | 23 | East |
| 11 | River Bogie | River Deveron | 5.54 | 0.011 | −16000 | Poor | No | 0 | 0 | 0 | 61.3 | 134 | 439 | 22 | East |
| 12 | River Dulnain | River Spey | 5.68 | 0.004 | −16000 | Good | No | 1 | 0 | 0 | 81.3 | 231 | 469 | 22 | East |
| 13 | River Feshie | River Spey | 5.7 | 0.006 | −10000 | Moderate | No | 1 | 0 | 0 | 121 | 350 | 508 | 23 | East |
| 14 | River Oich | River Ness | 5.17 | 0.032 | −10000 | Good | Yes | 0 | 0 | 0 | 50.8 | 31.4 | 484 | 22 | East |
| 15 | River Beauly | River Beauly | 5.25 | 0.019 | −12000 | Good | Yes | 0 | 0 | 0 | 3.72 | 9.1 | 450 | 21 | East |
| 16 | River Meig | River Conon | 5.2 | 0.032 | −10000 | Good | No | 0 | 0 | 0 | 29.8 | 118 | 469 | 24 | East |
| 17 | Abhainn a'Ghlinne Mhor | River Carron (Bonar Bridge) | 5.49 | 0.015 | −12000 | Poor | Yes | 0 | 0 | 0 | 23.3 | 210 | 451 | 24 | East |
| 18 | River Oykel | Oykel-Cassley-Shin System | 5.49 | 0.015 | −12000 | Good | No | 1 | 0 | 0 | 2.75 | 7.9 | 436 | 25 | East |
| 19 | River Shin | Oykel-Cassley-Shin System | 5 | 0.044 | −12000 | Good | Yes | 0 | 0 | 0 | 9.13 | 78 | 442 | 25 | East |
| 20 | River Brora | River Brora | 5.29 | 0.015 | −15000 | Good | Yes | 0 | 0 | 0 | 20.9 | 79.1 | 417 | 24 | East |
| 21 | River Helmsdale | River Helmdale | 5.49 | 0.009 | −16000 | Good | No | 0 | 0 | 0 | 5.68 | 20.2 | 393 | 24 | East |
| 22 | Strath Burn | Wick River | 5.4 | 0.024 | −16000 | Good | No | 0 | 0 | 0 | 15.3 | 28.9 | 382 | 24 | East |
| 23 | Glutt Water | River Thurso | 5.21 | 0.028 | −16000 | Good | No | 1 | 0 | 0 | 54.1 | 169 | 417 | 25 | West |
| 24 | River Borgie | River Borgie | 5.55 | 0.011 | −15000 | Good | No | 1 | 0 | 0 | 8.05 | 68.9 | 370 | 24 | West |
| 25 | River Dionard | River Dionard | 5.59 | 0.005 | −15000 | Good | No | 0 | 0 | 1 | 11 | 81.4 | 379 | 24 | West |

*(Continued)*

**Table 1.** (Continued)

| Site | Site Name | Drainage System | Rarefied $A_r$ | $F_{st}$ | Time Since Deglaciation | Water Quality | Modi-fied | Salmon Protection | Water-way Protection | Surround-ings Protection | Dis-tance to the Sea | Alti-tude | Con-nec-tivity | Tem-pera-ture | Coast |
|---|---|---|---|---|---|---|---|---|---|---|---|---|---|---|---|
| 26 | Abhainn Ghri-omarstaidh | Grimesta | 5.51 | 0.017 | −15000 | Good | No | 1 | 0 | 0 | 0.113 | 9 | 411 | 25 | West |
| 27 | Abhainn an Uisge | Steisavat | 5.54 | 0.014 | −15000 | Good | No | 0 | 0 | 0 | 3.18 | 9 | 399 | 22 | West |
| 28 | Abhainn Rodhag | Howmore River | 5.18 | 0.042 | −16000 | Good | No | 0 | 0 | 0 | 4.38 | 20.8 | 433 | 22 | West |
| 29 | River Laxford | River Laxford | 5.47 | 0.012 | −15000 | Good | No | 0 | 1 | 0 | 10.8 | 39.2 | 377 | 23 | West |
| 30 | River Polly | River Polly | 5.46 | 0.009 | −15000 | Good | No | 0 | 0 | 1 | 2.14 | 18 | 372 | 24 | West |
| 31 | River Kerry | River Kerry | 5.32 | 0.017 | −15000 | Good | Yes | 0 | 1 | 0 | 1.48 | 20.9 | 379 | 24 | West |
| 32 | River Torridon | River Torridon | 5.34 | 0.015 | −15000 | High | No | 0 | 0 | 1 | 7.15 | 86.8 | 401 | 22 | West |
| 33 | River Snizort | River Snizort | 5.34 | 0.024 | −16000 | Good | No | 0 | 0 | 0 | 7.08 | 77.8 | 409 | 24 | West |
| 34 | Glenmore River | Glenmore River | 5.41 | 0.026 | −15000 | Good | No | 0 | 0 | 0 | 8.14 | 46.4 | 401 | 23 | West |
| 35 | Strontian River | Strontian River | 4.79 | 0.075 | −10000 | Mod-erate | No | 0 | 0 | 1 | 3.08 | 17.8 | 453 | 25 | West |
| 36 | River Awe | River Awe | 5.06 | 0.059 | −12000 | Good | Yes | 0 | 0 | 0 | 1.14 | 7.2 | 461 | 22 | West |
| 37 | Allt a'Chaol Ghlinne | River Ruel | 5.68 | 0.011 | −12000 | Good | Yes | 1 | 0 | 0 | 30.8 | 38.3 | 603 | 19 | West |
| 38 | River Leven | River Leven (Dunbarton-shire) | 5.03 | 0.061 | −12000 | Mod-erate | Yes | 0 | 0 | 0 | 5.37 | 9 | 599 | 22 | West |
| 39 | Endrick Water | River Leven (Dunbarton-shire) | 4.66 | 0.084 | −12000 | Good | No | 0 | 0 | 0 | 58.5 | 148 | 651 | 22 | West |
| 40 | Allander Water | River Clyde | 5.36 | 0.025 | −12000 | Mod-erate | Yes | 0 | 0 | 0 | 14.7 | 35.8 | 609 | 22 | West |
| 41 | River Clyde | River Clyde | 5.41 | 0.026 | −15000 | Good | No | 0 | 0 | 0 | 33.4 | 34 | 640 | 22 | West |
| 42 | Dusk Water | River Garnock | 5.88 | 0.014 | −15000 | Mod-erate | No | 0 | 0 | 0 | 71.7 | 27.5 | 563 | 21 | West |
| 43 | River Ayr | River Ayr | 5.53 | 0.011 | −15000 | Good | No | 0 | 0 | 0 | 16.1 | 37.9 | 569 | 21 | West |
| 44 | River Stinchar | River Stinchar | 5.68 | 0.007 | −16000 | Good | No | 0 | 0 | 0 | 9.05 | 28 | 546 | 22 | West |
| 45 | Water of Luce | Water of Luce | 5.52 | 0.018 | −16000 | Good | No | 0 | 0 | 0 | 6.19 | 24 | 604 | 22 | West |
| 46 | Polharrow Burn | River Dee (Kircud-brightshire) | 5.26 | 0.03 | −12000 | Poor | Yes | 0 | 0 | 0 | 42.2 | 80.2 | 670 | 22 | West |
| 47 | River Nith | River Nith | 5.71 | 0.009 | −12000 | Mod-erate | No | 0 | 0 | 0 | 8.56 | 13.9 | 679 | 23 | West |
| 48 | River Annan | River Annan | 5.65 | 0.011 | −12000 | Poor | Yes | 0 | 0 | 0 | 33.3 | 33.3 | 708 | 24 | West |

Note: $A_r$, rarefied allelic richness.

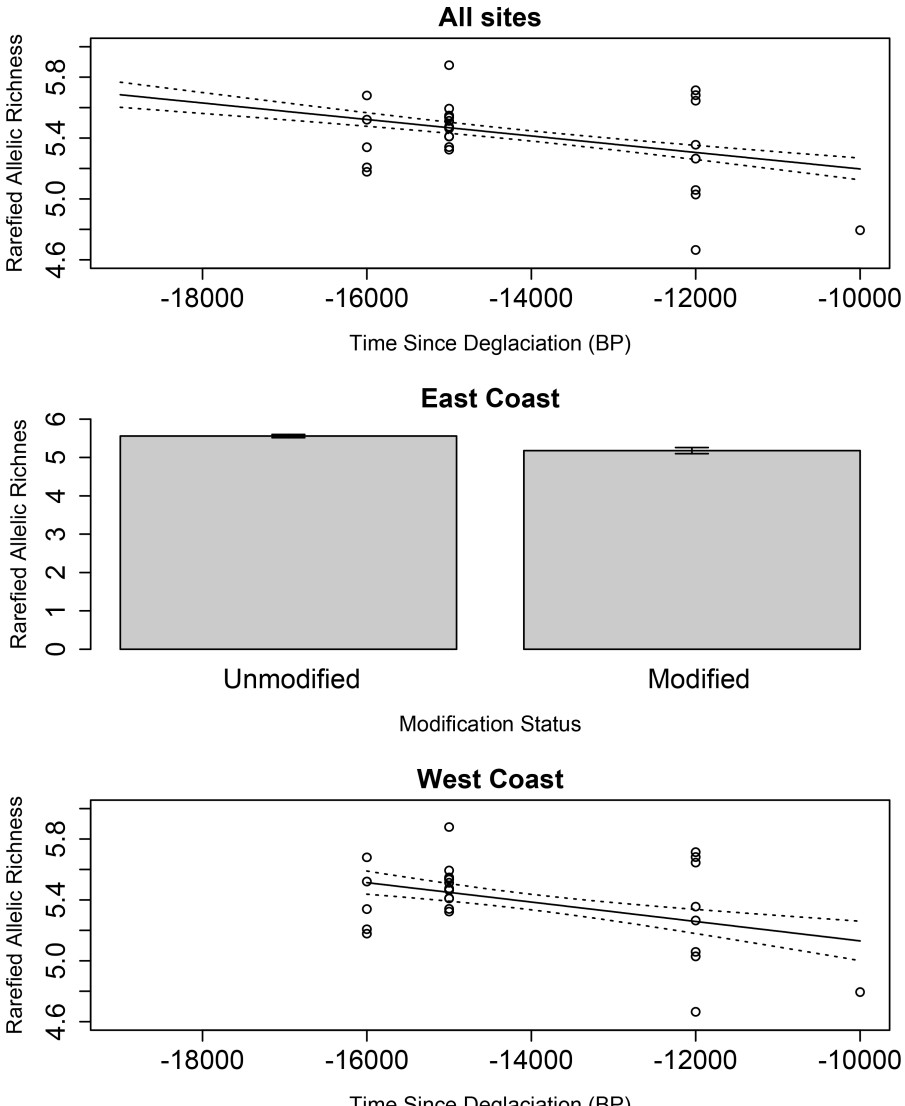

**Fig 2. Results from the best rarefied allelic richness GLMs as chosen by AICc with means and one standard error from the mean represented.**

deglaciation, which was also reported in Cauwelier *et al.* [5]. However, contemporary factors also seemed to be playing a role in shaping genetic variation, particularly when considering rivers draining on the east coast of Scotland, where waterbody modification had an effect on the same order of magnitude as time since deglaciation. Additionally, an effect of distance to sea was also detected but the effect size was very small.

## Post-pleistocene colonisation

Previous studies have shown that the founder effects associated with recolonisation of newly ice-free rivers [3–6,8] are still the predominant determinant of contemporary genetic structuring in *S. salar* populations. These results, showing low genetic diversity and high differentiation in the most recently deglaciated rivers, are akin to the stepping-stone model [50] and central-marginal hypothesis [51]. This result is likely to be independent of the type of molecular markers used, as SNP based analysis [52] resulted in very similar groupings to when microsatellites were used [5].

**Table 2. Summary of results from the best GLMs for each coast, explaining rarefied allelic richness patterns using backwards elimination.**

**All Sites**

| Predictors | Term df | Residual df | F-value | Adjusted R-squared | P-value |
|---|---|---|---|---|---|
| Time since Deglaciation | 1 | 46 | 12.4 | 0.195 | **< 0.001** |

**East Coast**

| Predictors | Term df | Residual df | F-value | Adjusted R-squared | P-value |
|---|---|---|---|---|---|
| Waterbody Modification | 1 | 20 | 22.6 | 0.507 | **< 0.001** |

**West Coast**

| Predictors | Term df | Residual df | F-value | Adjusted R-squared | P-value |
|---|---|---|---|---|---|
| Time Since deglaciation | 1 | 24 | 4.78 | 0.131 | **0.039** |

**Table 3. The sum of marginal posterior probabilities that include a given factor, produced by GESTE when all nine factors were included.**

| Factors | Sum of Marginal Posterior Probabilities | | |
|---|---|---|---|
| | All Sites | East Coast | West Coast |
| Time since deglaciation | 0.596 | 0.524 | 0.533 |
| Distance to the sea | 0.194 | 0.144 | 0.063 |
| Water quality | 0.053 | 0.044 | 0.069 |
| Waterbody modification | 0.04 | 0.162 | 0.063 |
| Connectivity | 0.029 | 0.035 | 0.057 |
| Protection of surroundings | 0.023 | N/A | 0.061 |
| Protection of *S. salar* | 0.022 | 0.044 | 0.064 |
| Protection of the waterbody | 0.022 | N/A | 0.045 |
| Temperature | 0.02 | 0.032 | 0.045 |

## Waterbody modification

A large effect of waterbody modification on genetic diversity and differentiation was only observed in the east coast analysis. The fact that genetic diversity on the east coast is significantly reduced at sites that have been heavily modified suggests, unsurprisingly, that there have been impediments to gene flow and stronger genetic drift as a result of these modifications. The non-negligible posterior probability of time since deglaciation and waterbody modification provided by GESTE further shows these two factors had an impact on the pattern of genetic differentiation seen today. Sites that have been deglaciated recently have been the major contributor to genetic differentiation, but waterbody modifications have likely reduced gene flow between populations and increased levels of genetic drift, increasing differentiation and reducing diversity further. This highlights how anthropogenic activities can have important effects on ecological timescales.

There were differences in the nature of the modifications between East and West river drainages, with hydro schemes dominating the modifications along the east coast. Modifications caused by hydro schemes vary in their nature, including, for example, both the storage of water behind dams and general water abstraction from run-of-river schemes. Meanwhile modifications for flood prevention and agriculture were more frequently found along the west coast. Indeed, hydro schemes were responsible for five of the six alterations on the east coast sites, whereas this is only true for four out of the seven modified sites on the west coast. The absence of an impact of waterbody modifications on diversity/differentiation along the west coast could suggest that these cases of flood prevention and irrigation have not significantly altered the genetic composition of these populations. Furthermore, the results provide another example that hydro schemes can have a marked impact on genetic structuring [21–23].

**Table 4. Posterior probabilities of the five most probable models for the site-specific GESTE analysis when considering five factors, at three different spatial scales.**

**All Sites**

| Model | Factors Included | Posterior Probability |
|---|---|---|
| 17 | Time since deglaciation | 0.549 |
| 21 | Time since deglaciation and distance to the sea | 0.296 |
| 18 | Time since deglaciation and water quality | 0.048 |
| 19 | Time since deglaciation and waterbody modification | 0.032 |
| 22 | Time since deglaciation, distance to the sea and water quality | 0.02 |

**East Coast**

| Model | Factors Included | Posterior Probability |
|---|---|---|
| 17e | Time since deglaciation | 0.405 |
| 25e | Time since deglaciation and distance to the sea | 0.171 |
| 19e | Time since deglaciation and waterbody modification | 0.155 |
| 21e | Time since deglaciation and protection of *S. salar* | 0.063 |
| 3e | Waterbody modification | 0.047 |

**West Coast**

| Model | Factors Included | Posterior Probability |
|---|---|---|
| 1w | Null | 0.43 |
| 17w | Time since deglaciation | 0.318 |
| 9w | Distance to the sea | 0.041 |
| 3w | Waterbody modification | 0.038 |
| 2w | Water Quality | 0.031 |

## Distance to the sea

Distance to the sea had a non-negligible posterior probability (all sites and east coast) in the GESTE analysis. However, the slopes (Fig 3) were very close to zero, which would suggest that the distance from the sea was not biologically meaningful. In both cases, the influence of the distance to the sea was accompanied by the time since deglaciation, which was unsurprising, given that colonisation might majorly have been in an upstream direction, as the ice sheet was receding. However, the results suggest that genetic differentiation decreased with increasing distance to the sea; the opposite of what previous studies have found [17-53].

The unexpected negative relationship between differentiation and distance to sea could be due to outlier sites. These outlier sites were characterised by high population-specific $F_{st}$ close to the mouth of the river or low $F_{st}$ at large distances from the sea. Coincidentally, the outlier sites were among the first (low $F_{st}$) or the most recent (high $F_{st}$) to become ice free.

Overall, the existence of such outliers suggests that the influence of the distance to the sea on genetic differentiation is overridden by the effect of time since deglaciation. When the outliers were removed, the relationship between differentiation and distance to the sea became positive. Although not statistically significant ($r=0.237$, $p=0.159$), this is more in line with the previous findings [17-53].

## Other factors

The other factors considered in the model which are influenced by anthropogenic activities (water quality, temperature and the various environmental protections) appeared to have had a minor impact on genetic structuring in these populations.

Contrasting results have been reported on the effect of temperature on genetic differentiation. Dionne *et al.* [17] showed that increases in genetic differentiation were associated with higher temperatures in some *S. salar* populations.

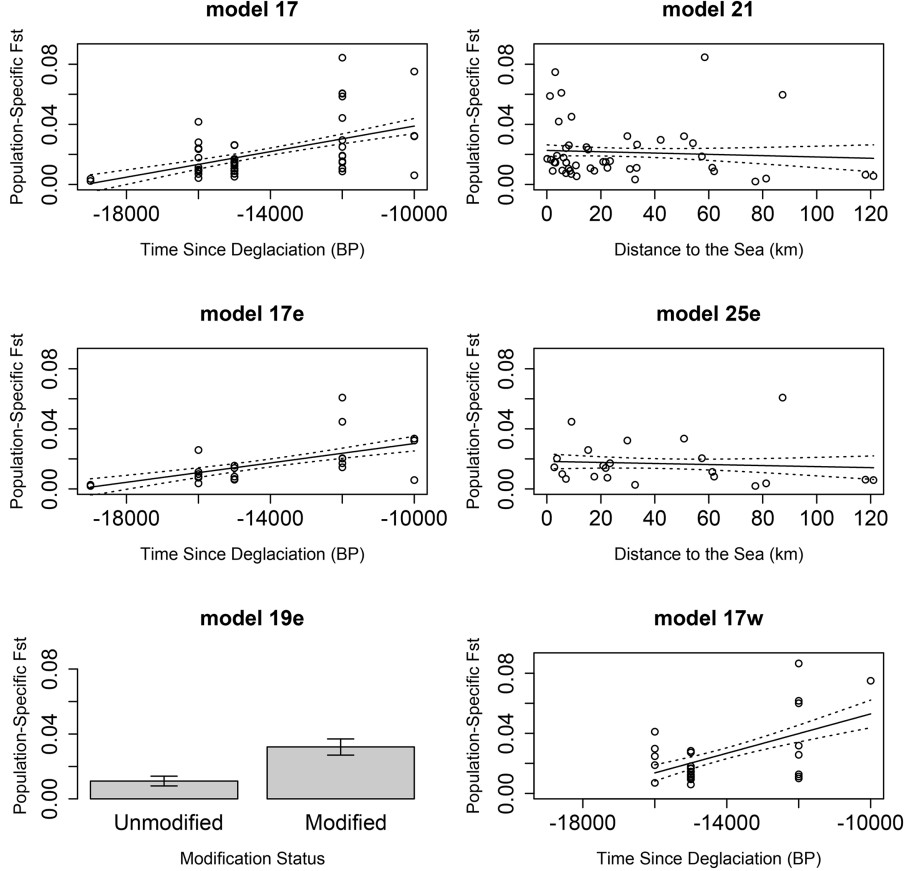

**Fig 3. Results from all the GESTE models with a non-negligible posterior probability with means and one standard error from the mean represented.** Model numbers relate to the numbers in Table 4.

In contrast, anadromous rainbow (steelhead) trout (*Oncorhynchus mykiss*) showed reduced differentiation with increased temperature [54]. This would suggest that more research is needed into the effect of temperature on population structuring in various species of Salmonids.

There were other environmental variables left unincluded due to a paucity of data and/or the level of complexity, such as those relating to hydrology and geography. For example, populations in close proximity to one another but located across geological boundaries have been shown to be more differentiated than populations within them, which might be linked to accuracy of homing [55].

### Considerations for the west coast

Similar to the analysis of all sites and the east coast sites, the results from the west coast analysis showed an effect of timing since deglaciation on genetic diversity, with lower diversity found in the most recently deglaciated rivers. It is, however, notable that the $R^2$ more than doubles if west coast sites are not included. It might be that the limited variability of the various factors considered played a role. For example, distance to sea varied from 0.113 to 58.5 km (mean = 14.9 and SD = 16.5) along the west coast, whilst this was between 2.75 and 121 km (mean = 42.2 and SD = 36.3) along the east coast. However, the low $R^2$, together with the finding that the null model was the best when looking at population-specific

$F_{st}$, might also suggest that, despite including many factors in this study, some other, more prominent factors, were missed.

One factor that was not considered in the model was the presence of aquaculture. Indeed, fish farms are almost exclusively located on the west coast of Scotland. Given that Gilbey *et al.* [56] have detected introgression from farmed fish in wild west coast populations, it cannot be ruled out that aquaculture has had a significant impact on the populations in this study. Glover *et al.* [27] showed that, in Norway, the presence and growth of the aquaculture industry has led to an erosion of wild *S. salar* population structure over time, with contemporary populations being less differentiated than their historical counterparts. However, the impact of aquaculture on wild populations varies across regions [25,56] and is very complex to measure and model [57,58]. As such, it would be very difficult to disentangle and robustly attribute the impact of aquaculture from the other local environmental characteristics of the rivers in question.

## Conclusions

In summary, the results from this study have reinforced the findings of Cauwelier *et al.* [5] but put the influence of post-Ice Age colonisation in the context of the other contemporary forces considered and differences in the type of anthropogenic perturbations between east and west coast river-drainage systems in Scotland. In this regard, the most important contemporary factor is waterbody modification affecting rivers draining in the east. However, other river characteristics and anthropogenic factors not considered in this study could also be important and these should be explored in more detail so as to account for the genetic variation not explained in this study.

## Supporting information

**S1 Table. Posterior means, modes and highest posterior density intervals (HPDI) for the best GESTE models.**
(DOCX)

**S1 File. Genetic data: Microsatellite data in GENALEX format.**
(XLSX)

## Acknowledgments

Thanks to Widar Wennevik for helpful comments and discussion. Thanks also to John Gilbey for assisting in the selection of the sample sites and to Synne Winterthun for her work on screening and QC-ing the samples for the additional microsatellites. This work was made possible due to the partial sample screening carried out under the Research Council of Norway project 280308 (SeaSalar).

## Author contributions

**Conceptualization:** Finn Cowell, Oscar E. Gaggiotti, Eef Cauwelier.

**Data curation:** Finn Cowell, Eef Cauwelier.

**Formal analysis:** Finn Cowell.

**Methodology:** Finn Cowell, Oscar E. Gaggiotti, Eef Cauwelier.

**Project administration:** Finn Cowell.

**Visualization:** Finn Cowell, Eef Cauwelier.

**Writing – original draft:** Finn Cowell.

**Writing – review & editing:** Finn Cowell, Oscar E. Gaggiotti, Eef Cauwelier.

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
