## [Decision Letter · Decision Letter 0]

4 Dec 2024

Dear Dr. Cowell,

Thank you for submitting your manuscript to PLOS ONE. After careful consideration, we feel that it has merit but does not fully meet PLOS ONE’s publication criteria as it currently stands. Therefore, we invite you to submit a revised version of the manuscript that addresses the points raised during the review process.

We look forward to receiving your revised manuscript.

Kind regards,

Sayyed Mohammad Hadi Alavi

Academic Editor

PLOS ONE

Additional Editor Comments:

It is my pleasure to inform you that the review process of your MS has completed. I was glad that two expert in the field of fish population genetics accepted our invitation to review your submission. They found your research very interesting, however had some concerns regarding the methods (particularly genomic analysis and and variables) to investigate population differentiation and diversity of Atlantic salmon in Scotland. Here, I would like to ask you revising your MS with careful consideration to the comments provided by reviewers, and provide them with point-by-point responses to the comments. Considering the comments, your revision will be subjected to another peer-review.

Reviewers' comments:

Reviewer's Responses to Questions

**Comments to the Author**

1. Is the manuscript technically sound, and do the data support the conclusions?

Reviewer #1: Partly

Reviewer #2: No

2. Has the statistical analysis been performed appropriately and rigorously?

Reviewer #1: No

Reviewer #2: No

3. Have the authors made all data underlying the findings in their manuscript fully available?

Reviewer #1: Yes

Reviewer #2: Yes

4. Is the manuscript presented in an intelligible fashion and written in standard English?

Reviewer #1: Yes

Reviewer #2: Yes

Reviewer #1: I have now read the manuscript “Post-Pleistocene Colonisation Rather than the Contemporary Environment has Most Influenced the Current Population Structure of Scottish Atlantic Salmon (Salmo salar)” In this study, the authors expand on a microsatellite panel for a subset of Atlantic Salmon populations previously studied in Scotland. They conclude that the Time since deglaciation is the major determinant for diversity and differentiation in those populations.

I want to start my review by stating that I have never directly worked with microsatellite data, as my expertise is leaning toward genomic analyses (both GBS and low coverage WGS). That being said, I was excited to see this enormous dataset revisited with such interesting questions as the determinants (historical vs contemporary) of genetic diversity. However, I wonder if this was the best choice to reduce the number of populations to increase the number of markers, as the near-exhaustive sampling of Scottish salmon populations in Cauwelier et al. 2018 was very impressive and could have yielded interesting insight into those questions by controlling for within and between watershed effects. However, I understand that it is unreasonable to ask the authors to revisit this design choice.

Rather, my main concern about this manuscript is the treatment of population differentiation, which is central to the study. In the current version, differentiation is treated as an average of the pairwise Fst of a focal population with all others present in the dataset. To me, this measure has little biological meaning and is very dependent on the populations included in the dataset.

As such, I believe the authors should consider reworking considerably the portions of their study related to differentiation. There is abundant literature on isolation-by-distance driving patterns of differentiation in salmonids, and right now I feel neither connectivity nor distance by sea is properly accounting for this. I suggest that the differentiation should be analyzed using pairwise Fst (by pair of populations), rather than ‘population-specific Fst’. This way, the author could introduce either the marine distance between river mouths or the total distance between sampling sites to first control for isolation-by-distance. Then, additional variables, such as the difference in timing of deglaciation between the 2 focal populations could be introduced as covariable, i.e. to test if two populations are more similar than expected given the geographical distance separating them if they occupy habitats that became free of ice at the same time. Other covariables could be used in a similar fashion.

In light of this, my assessment of the present manuscript is that it needs a major revision of the central analyses in order to yield insight in the questions proposed by the authors. I have attached below some more specific comments.

METHODS

Ln 175: It is rather unclear what Connectivity represents in this context, especially since your dataset does not pretend to encompass every salmon population in the region, as you specifically used a portion of those studied in Cauwelier et al. 2018. As used presently, the measure would be more meaningful if estimated using every known Scottish population, but even then, I feel like the pairwise river+marine distances would be better used in an isolation-by-distance analysis, as I discussed above.

RESULTS

Ln 215-216: Is this positive or negative? What does this mean concerning the filtering? That no markers were removed?

Suggestion: As Hardy-Weinberg equilibrium deviation and positive tests for LD were not significantly greater than the number expected due to effects of chance alone, all 33 microsatellite loci were analyzed.

Ln 216: Suggestion for clarity: Average rarefied allelic richness across the 48 Scottish Atlantic Salmon populations equaled 5.47 (SD = 0.25).

Ln 218: Add the population number (in parentheses) as in Fig. 1 to the river name, as this is much easier to follow.

Ln 239: This whole sentence lacks in readability. It should also be clear that we moved on from diversity to differentiation.

Ln 242: ‘Non-negligeable’ : Did you use a specific threshold?

DISCUSSION

Ln 300-301: Could you please provide reference for this claim? It seems to be very specific to anadromous fish in this specific landscape.

Ln 304-306: Assuming we are talking about anadromous populations of salmons, I’m not sure I would refer to dispersal inside a watershed as colonization. Can we really expect a strong founder effect as fish that migrate along a river start spawning in newly available habitats inside their natal watershed? I wonder, but most generally I feel that the implications of the anadromy inside the studied populations are not sufficiently discussed in this paper, both in the introduction and discussion.

FIGURES

Fig. 1: Could you add the genetic clustering information referred to in the text (Cauwelier et al. 2018), or at the very least the membership to either the West or East group relevant to the hierarchical analysis?

Fig. 2: This is rather stylistic note, but I feel the trends would be more apparent and comparable if the plots were closer to a square in shape, as there is not a lot of density of points to show on the x-axis. That would also allow the three panels to by side-by-side, so we can better appreciate the differences in allelic richness in the west and east. For the same reason, I also suggest that you use the same extent (min and max) on both the x- and y-axis for all three plots.

This would also allow you to add a second line of plots to show the effect of distance to sea (all sites + east), which I don’t understand the absence in Figures. This would help to support the claims made at Ln 278-279.

Fig. 3: Similar to my comment on Fig. 2, I suggest that the panels be reorganised so the reader can more visually compare the results of different model and region.

Reviewer #2: See attachment.

PONE-D-24-37086

“Post-Pleistocene Colonisation Rather than the Contemporary Environment has Most Influenced the Current Population Structure of Scottish Atlantic Salmon”

The manuscript is an interesting follow-up to Cauwelier et al. (2018). The submitted manuscript attempts to expand on the previous report on Atlantic salmon population structure in Scotland. I can understand why the authors invested time and resources into the new study.

1) Unfortunately, the metrics selected to study simply don’t have enough variability to build meaningful models. Two metrics are class/categorical variables. The temperature data varies by only a few degrees. The river length were short in terms of distance salmon travel to reach spawning sites; most of these distances could be travelled in a day or two and therefore it is not a real measure of a distance challenge that might be a predictor of population structure. As an aside, the authors don’t show what appears to be nine (9) metrics in a list or summary table. Related to this is the metric “elevation” which is mentioned but does appear again; river slope might be useful metric to add.

2) The distance/connectivity is problematic, “population-specific measure of connectivity”, because distance is distance. By combining these two measures, the larger number (between rivers) most probably overrides the biological relevance of the in-river distance.

3) Selecting 48 of the original salmon rivers was described as a random process with no further clarification. The original study demonstrated that the overall group of Scottish salmon is partitioned into several definable units. The authors don’t mention how you addressed the potential impacts of a simple random selection from classes of data.

4) There are couple of editorial comments in the manuscript you may find useful for future writing. For example, melting ice sheets (glaciers) aren’t actually moving, just melting. If you were invoking the movement effect as a metric of interest, then you would need to demonstrate that glaciers were surging and concurrently varying accessibility by salmon to the rivers.

Overall, the problem with the lack of variability within metrics is demonstrated in the statistically, non-significant models. A different set of metrics, each with reasonable variability among sites might help to tease out the role of the contemporary environment.

**Do you want your identity to be public for this peer review?** For information about this choice, including consent withdrawal, please see our Privacy Policy

Reviewer #1: No

Reviewer #2: No

---

## [Author Response · Author response to Decision Letter 1]

10 Jan 2025

Response:

We are grateful for the effort on the part of our reviewers to provide us with comments, some of which are extremely helpful. We have accepted many of their recommendations, such as making changes to figures and making stylistic changes to the writing, improving the clarity of the manuscript. There are however some major issues that both reviewers took with regard to our methodology. We disagree with the premise of these criticisms and address them fully.

5. Review Comments to the Author

Reviewer #1: I have now read the manuscript “Post-Pleistocene Colonisation Rather than the Contemporary Environment has Most Influenced the Current Population Structure of Scottish Atlantic Salmon (Salmo salar)” In this study, the authors expand on a microsatellite panel for a subset of Atlantic Salmon populations previously studied in Scotland. They conclude that the Time since deglaciation is the major determinant for diversity and differentiation in those populations.

I want to start my review by stating that I have never directly worked with microsatellite data, as my expertise is leaning toward genomic analyses (both GBS and low coverage WGS). That being said, I was excited to see this enormous dataset revisited with such interesting questions as the determinants (historical vs contemporary) of genetic diversity. However, I wonder if this was the best choice to reduce the number of populations to increase the number of markers, as the near-exhaustive sampling of Scottish salmon populations in Cauwelier et al. 2018 was very impressive and could have yielded interesting insight into those questions by controlling for within and between watershed effects. However, I understand that it is unreasonable to ask the authors to revisit this design choice.

Rather, my main concern about this manuscript is the treatment of population differentiation, which is central to the study. In the current version, differentiation is treated as an average of the pairwise Fst of a focal population with all others present in the dataset. To me, this measure has little biological meaning and is very dependent on the populations included in the dataset.

As such, I believe the authors should consider reworking considerably the portions of their study related to differentiation. There is abundant literature on isolation-by-distance driving patterns of differentiation in salmonids, and right now I feel neither connectivity nor distance by sea is properly accounting for this. I suggest that the differentiation should be analyzed using pairwise Fst (by pair of populations), rather than ‘population-specific Fst’. This way, the author could introduce either the marine distance between river mouths or the total distance between sampling sites to first control for isolation-by-distance. Then, additional variables, such as the difference in timing of deglaciation between the 2 focal populations could be introduced as covariable, i.e. to test if two populations are more similar than expected given the geographical distance separating them if they occupy habitats that became free of ice at the same time. Other covariables could be used in a similar fashion.

In light of this, my assessment of the present manuscript is that it needs a major revision of the central analyses in order to yield insight in the questions proposed by the authors. I have attached below some more specific comments.

Response:

We are grateful for many of this reviewer’s comments and the time they clearly put into reading our manuscript. While we may not be able to allay all of their misgivings, we believe that changes to our manuscript will be helpful to future readers.

This reviewer believes that the population-specific measures of Fst used in this study have little biological meaning. We strongly disagree with this idea as population-specific Fst is strongly grounded in fundamental concepts in metapopulation biology and population genetics. Moreover, it can be argued that, despite being widely used, pairwise Fst measures are ad-hoc and lack theoretical grounding. The study of isolation by distance using pairwise Fst relies on two completely unrealistic assumptions:

1) Differentiation between a pair of populations is not influenced by the exchange of migrants that the two focal populations may have with other local populations. In other words, the fact that differentiation between a pair of populations can be influenced by exchange of genetic material those two populations have with other local populations can be completely ignored.

2) Differences in local population size has no effect of genetic differentiation between the pair of populations. This assumption ignores the fact that differences in population size also determine differences on the effect of genetic drift on genetic differentiation.

Clearly these two assumptions contradict standard population genetic principles stating that both genetic drift and migration influence genetic differentiation in a spatially-structured population.

On the other hand, the population-specific Fst concept explicitly acknowledges the fact that both global patterns of migration as well as local population size influence genetic differentiation between populations, as explicitly stated by population genetics theoretical principles that can be found in any introductory population genetics textbook. Moreover, it is strongly grounded on the concept of population connectivity, as developed in the domain of metapopulation biology. Indeed, any meaningful measure of isolation needs to take into account both distance and size of the local populations (see Box 11.1 in Hanski 1999; for a more advanced treatment of this issue see Hanski & Gaggiotti 2004).

All the above theoretically- and conceptually-grounded arguments justify the use of local population Fst while disqualifying the use of pairwise Fst. We need to point out that the fact that a measure or statistic is widely used does not imply that it is appropriate or biologically meaningful.

In addition to the theoretical support for population-specific measures of Fst, and its ability to detect any existing IBD, while considering global migration patterns and differences in local population size, we are also sure that an IBD signal would not in fact be detected using pairwise measures of Fst and distance in this instance. This is because we had previously conducted a classic mantel test using these metrics. The result of this test corroborated our result using population-specific measures. We have now included a description of the mantel test methodology (lines 211-215) and the associated result (lines 263-264). It is however worth pointing out that we prefer the use of population-specific variables as supported by the theoretically-grounded arguments we outline above.

Reviewer #1

METHODS

Ln 175: It is rather unclear what Connectivity represents in this context, especially since your dataset does not pretend to encompass every salmon population in the region, as you specifically used a portion of those studied in Cauwelier et al. 2018. As used presently, the measure would be more meaningful if estimated using every known Scottish population, but even then, I feel like the pairwise river+marine distances would be better used in an isolation-by-distance analysis, as I discussed above.

Response:

Our study uses connectivity between sites to investigate if such a pattern is present between our sampling populations. The sites used in this study are a subset of those used in Cauwelier et al. (2018) and these encompass the major clustering groups identified in 2018, giving us good geographical coverage as a result. We therefore feel confident in using connectivity (which we clearly describe in lines 178-180 as the mean of all pairwise distances between a population and all other populations) as a measure. This is a method that has been applied before in highly cited publications (e.g., Foll & Gaggiotti, 2006; Gaggiotti et al., 2009; Hand et al., 2016; Kittlein & Gaggiotti, 2008). In addition, we have now included the mantel test results, corroborating our population-specific results (lines 263-264).

Reviewer #1

RESULTS

Ln 215-216: Is this positive or negative? What does this mean concerning the filtering? That no markers were removed?

Suggestion: As Hardy-Weinberg equilibrium deviation and positive tests for LD were not significantly greater than the number expected due to effects of chance alone, all 33 microsatellite loci were analyzed.

Response:

Done.

Reviewer #1

Ln 216: Suggestion for clarity: Average rarefied allelic richness across the 48 Scottish Atlantic Salmon populations equaled 5.47 (SD = 0.25).

Response:

Done, except that we retain the word mean, rather than use the word average.

Reviewer #1

Ln 218: Add the population number (in parentheses) as in Fig. 1 to the river name, as this is much easier to follow.

Response:

Done.

Reviewer #1

Ln 239: This whole sentence lacks in readability. It should also be clear that we moved on from diversity to differentiation.

Response:

We have changed this sentence so that it is now clear that we are talking about genetic differentiation (lines 258-260).

Reviewer #1

Ln 242: ‘Non-negligeable’ : Did you use a specific threshold?

Response:

No, this was very much an interpretation based on the fact that the marginal posterior probabilities were 0.524, 0.162 and 0.144 for time since deglaciation, waterbody modification and distance to the sea, respectively. The next highest was 0.044 and we considered this to be low enough not to discuss in detail, but we do not of course consider these variables to have no biological meaning, only that their effect on genetic differentiation is quite small. One of the advantages of using a Bayesian analysis is that it allows us to consider more nuance in the results by not using fixed thresholds. In order to clarify this, we have changed the wording from “non-negligible” to “relatively high” (line 262).

Reviewer #1

DISCUSSION

Ln 300-301: Could you please provide reference for this claim? It seems to be very specific to anadromous fish in this specific landscape.

Response:

This result has been found in two separate studies on different parts of the east coast of Canada (Dionne et al., 2008; Harris et al., 2014). These references are cited in the previous sentence, in the previous paragraph. We take this opportunity to rewrite this sentence explaining why one might expect a positive relationship between genetic differentiation and distance to the sea (lines 322-324).

Reviewer #1

Ln 304-306: Assuming we are talking about anadromous populations of salmons, I’m not sure I would refer to dispersal inside a watershed as colonization. Can we really expect a strong founder effect as fish that migrate along a river start spawning in newly available habitats inside their natal watershed? I wonder, but most generally I feel that the implications of the anadromy inside the studied populations are not sufficiently discussed in this paper, both in the introduction and discussion.

Response:

The time between the lower section of the watershed and the upper sections of the watershed becoming ice free numbers in thousands of years. If colonisation occurred downstream, this would also be expected to be a lengthy process owing to the slow nature of isostatic rebound. We therefore feel that the use of the term colonisation is justified.

Reviewer #1

FIGURES

Fig. 1: Could you add the genetic clustering information referred to in the text (Cauwelier et al. 2018), or at the very least the membership to either the West or East group relevant to the hierarchical analysis?

Response:

Done.

Reviewer #1

Fig. 2: This is rather stylistic note, but I feel the trends would be more apparent and comparable if the plots were closer to a square in shape, as there is not a lot of density of points to show on the x-axis. That would also allow the three panels to by side-by-side, so we can better appreciate the differences in allelic richness in the west and east. For the same reason, I also suggest that you use the same extent (min and max) on both the x- and y-axis for all three plots.

Response:

Done.

Reviewer #1

This would also allow you to add a second line of plots to show the effect of distance to sea (all sites + east), which I don’t understand the absence in Figures. This would help to support the claims made at Ln 278-279.

Response:

For our genetic diversity analysis, using rarefied allelic richness, we used a frequentist approach. The effect of distance to the sea was not statistically significant (see table 1) and therefore we did not include it in the plots. The model does however contain distance to the sea as a term in the model (as chosen by AIC) and therefore the plots in figure 2 show the results of the significant variables, accounting for the effect of distance to the sea. We have added an extra sentence into the figure legend in order to clarify this.

Reviewer #1

Fig. 3: Similar to my comment on Fig. 2, I suggest that the panels be reorganised so the reader can more visually compare the results of different model and region.

Response:

Done

Reviewer #2: See attachment.

PONE-D-24-37086

“Post-Pleistocene Colonisation Rather than the Contemporary Environment has Most Influenced the Current Population Structure of Scottish Atlantic Salmon”

The manuscript is an interesting follow-up to Cauwelier et al. (2018). The submitted manuscript attempts to expand on the previous report on Atlantic salmon population structure in Scotland. I can understand why the authors invested time and resources into the new study.

Response:

We are grateful for this reviewer for their helpful comments. I believe our manuscript now reads more clearly as a result. The reviewer clearly has several misgivings regarding the data that we used and we address this issue in our response.

1) Unfortunately, the metrics selected to study simply don’t have enough variability to build meaningful models. Two metrics are class/categorical variables. The temperature data varies by only a few degrees. The river length were short in terms of distance salmon travel to reach spawning sites; most of these distances could be travelled in a day or two and therefore it is not a real measure of a distance challenge that might be a predictor of population structure. As an aside, the authors don’t show what appears to be nine (9) metrics in a list or summary table. Related to this is the metric “elevation” which is mentioned but does appear again; river slope might be useful metric to add.

Response:

This reviewer takes issue with an apparent lack of variability in some of the metrics we used. They note our use of categorical variables but do not explain why this results in a lack of variability. We acknowledge in our manuscript that the use of these variables has limitations (as do all data) but we feel the use of standardised metrics of water quality and waterbody modification are justified in their use in this study. The reviewer takes further issue with our use of temperature data that they have decided does not vary enough. While we can certainly imagine circumstances under which higher temperatures could have a greater impact than the temperatures in the rivers we are studying, we are using actually existing data so our conclusions are that the temperatures that our rivers have experienced are not having a major impact. Given the data, this is a valid conclusion and does not say anything about, for example, if temperatures were to rise further due to our climate change.

A major misunderstanding seems to have taken place regarding the distance to the sea measures we used. Firstly, Scottish rivers are only so long, meaning that, when investigating distance to the sea effects in Scotland, it is a simple reality that distances will be shorter compared to other parts of the world. Within Scotland, our sites certainly inc

---

## [Decision Letter · Decision Letter 1]

8 Apr 2025

Dear Dr. Cowell,

Thank you for submitting your manuscript to PLOS ONE. After careful consideration, we feel that it has merit but does not fully meet PLOS ONE’s publication criteria as it currently stands. Therefore, we invite you to submit a revised version of the manuscript that addresses the points raised during the review process.

We look forward to receiving your revised manuscript.

Kind regards,

Sayyed Mohammad Hadi Alavi

Academic Editor

PLOS ONE

Additional Editor Comments:

Dear Finn Cowell; The review of revised MS you submitted to PLOS ONE has completed. To reach the current decision, we invited additional reviewers to review the revised MS. Referees provided us with very valuable comments, I warmly appreciate for their agreement to review this submission and for very valuable comments. I would like to ask you revising your MS for another time according to the comments appended below. In addition to the referees’ comments, please consider those of mine in your revision:

1- I agree with R#4 to re-locate Table S1 in the main body of the paper.

2- Describe the methods in detail.

3- Follow the guide from PLOS ONE for the use of map: it needs to indicate copyright.

4- For figures 2 and 3: Increase the font size for numbers shown in X-axis and Y-axis as well as title for each panel.

Reviewers' comments:

Reviewer's Responses to Questions

**Comments to the Author**

Reviewer #1: All comments have been addressed

Reviewer #2: (No Response)

Reviewer #3: (No Response)

Reviewer #4: (No Response)

2. Is the manuscript technically sound, and do the data support the conclusions?

Reviewer #1: Yes

Reviewer #2: No

Reviewer #3: Yes

Reviewer #4: Yes

3. Has the statistical analysis been performed appropriately and rigorously?

Reviewer #1: Yes

Reviewer #2: Yes

Reviewer #3: Yes

Reviewer #4: Yes

4. Have the authors made all data underlying the findings in their manuscript fully available?

Reviewer #1: Yes

Reviewer #2: Yes

Reviewer #3: Yes

Reviewer #4: Yes

5. Is the manuscript presented in an intelligible fashion and written in standard English?

Reviewer #1: Yes

Reviewer #2: Yes

Reviewer #3: Yes

Reviewer #4: Yes

Reviewer #1: I enjoyed reading the manuscript for a second time, with a bit more perspective. As stated in my original review, I was less familiar with some of the methods and approaches used here, so I apologize to the authors if my original review lacked nuance at times. I want to thank them for their enlightening, if not humbling, response.

I consider this study to offer a valuable contribution to the field, and now that my concerns have been addressed, I recommend its publication in its current form.

Reviewer #2: I have enclosed more comments in two files. One is a response to your response to my review and the other provides some additional commentary on your manuscript (I used the section with your original manuscript and your responses to the reviews). My fundamental criticism was that your selected metrics don’t have the variability necessary to tease out the mechanisms of population structure. Your overall goal and the methods are not at issue. You have a great looking genetics data set and I am confident that environment metrics exist that can help delineate mechanisms of structuring.

Reviewer #3: PONE-D-24-37086

“Post-Pleistocene Colonisation Rather than the Contemporary Environment has Most

Influenced the Current Population Structure of Scottish Atlantic Salmon”

I have read the manuscript thoroughly; the authors are trying to explain the impacts of deglaciation compared to contemporary environmental changes using Cauwelier et al. (2018) microsatellite data. I also read the previous revisions by 2 different reviewers and the comprehensive responses of authors to the comments. I do not know the background of two previous reviewers but from my point of view, the subject is interesting and the methodology used is brilliant and correct based on the available data. The results and corresponding discussion also seem logical to me. The structure of the manuscript is strong enough after going through two revisions. Finally, I believe the manuscript is full-fits enough and can be published as it is.

Reviewer #4: In this manuscript, the authors attempt to show the relative contributions of historical and contemporary factors affecting genetic diversity and differentiation of Atlantic salmon in Scotland using microsatellite genotypes from populations previously analyzed in Cauwelier et al. (2018). Previous reviewers had called into question some of the methodology used in this manuscript, but I largely agreed with the authors in response to most of these concerns, outlined specifically below. I do, however, think that while the GLMs and GESTE models are valid methods of analyzing this data, the inferences that can be drawn from them are relatively weak, and there are additional methods and alternative approaches that could be used to supplement the authors’ findings and make the takeaways of the manuscript more convincing. Additionally, I think the writing style lacks specificity in key areas of the introduction, methods and discussion, and as a result parts of the manuscript are difficult to interpret. Overall, this manuscript focuses on clear and relevant ecological questions using an expansive microsatellite dataset, and I would recommend this manuscript for publication with a few minor improvements.

Major Issues

While I do not share the second reviewer’s concerns about the lack of variability in the environmental data, I do think that the results of the paper are somewhat difficult to interpret because how the environmental variation is spread across the sampled populations is not well described. Table S1 should be moved into the main text of the paper and the authors could consider additional map figures showing the most relevant environmental information for each of the sampled populations.

I agree with the authors that the use of population-specific Fst is absolutely appropriate for these analyses. While I could see some utility of comparing isolation by distance tests with tests of isolation by resistance/isolation by environment using pairwise Fst and/or other genetic distance metrics, because of the coarseness of many of the environmental variables used, I do not think that those tests would be very informative, at least for the goals of this manuscript.

There are other approaches that could be included to strengthen the inferences drawn in the manuscript. Redundancy analysis would be a relatively simple addition to assess the relative importance of environmental factors on genetic variation that could be directly compared to the results from the GLMs and GESTE, as well as show if there are any markers and populations are driving patterns associated with each environmental variable. Coalescence and demographic reconstruction analyses could also be relevant inclusions that could corroborate the timing of glacial retreats being the major factor structuring salmon populations in the study area.

One of the main conclusions of the paper is that contemporary habitat modifications are affecting genetic connectivity of these populations, but the effect size detected by GESTE is small, and analysis of genetic differentiation alone is not likely to demonstrate a strong signal of fragmentation due to relatively recent habitat modifications. This claim in particular would benefit from an analysis comparing patterns in historical gene flow (e.g. using a coalescence-based modelling approach like MIGRATE) versus contemporary gene flow (i.e. methods based on identifying migrants and their offspring in the sampled individuals), and seeing if populations in systems that have had habitat modifications show drastically different historical vs. contemporary patterns of connectivity.

Minor comments:

Abstract

Lines 35-37: Be more specific, what were the effects of waterbody modification? You should also mention what metrics you looked at in the abstract.

Introduction

Lines 45-52: This paragraph feels unfocused and does not introduce what this manuscript is about. It might be better to focus on how the environment and demographic processes interact to produce patterns in genetic variation before moving into how these processes tend to manifest in salmon populations specifically. The general biological information about salmonids that affects population genetic processes is important background information, but would be more appropriate for later in the introduction.

Lines 78-100: The previous section of the introduction laying out the natural history of glaciation and the impact on salmonid populations was quite effective at providing context, whereas this section reads more as a point-by-point justification for the environmental variables you included. This section would be more effective if it were reorganized into more of a narrative format like the above section was presented, giving a history of how anthropogenic factors have altered the contemporary landscape affecting these populations.

Methods

Lines161-163: It would be good to include a short description of the factors used to classify water quality.

Line163: This brief sentence is the only mention of waterbody modification in the methods. It is very important to describe what how this variable is constructed, especially since it is the most important contemporary effect you discuss later the results and discussion, but readers have no context as to what exactly this variable represents.

Lines 169-171: It is not clear what the difference is between dummy variables 2 and 3. Does 3 include only riparian buffers or any terrestrial area in the watershed? What are the regulations designed to protect salmon versus protective regulations not designed for salmon? Without any details, these variables seem like they could encompass a very wide variety of regulations that would have highly variable effects on salmonid populations and it is hard for readers to assess what the dummy variables are representing.

Line 177: What is the temporal resolution of this estimate? Is this a temperature that the fish might have experienced for an hour? A day? A month? Choosing the maximum also seems like it would be subject to outlier events that may not reflect the general thermal suitability of a stream. How closely does the max temp co-vary with the average temp?

Lines 177-178: Unclear how temperature is linked to the time since deglaciation.

Lines 194: Specify that rarefied allelic richness was the diversity measure used as the dependent variable in the GLMs

Lines 204-205: The pilot runs are used as the burn-in period for GESTE. You wouldn’t have a thinning interval for a burn-in period.

Lines 219-223: Was there hierarchical structure detected in the Cauwelier et al. 2018 paper? If the east and west coasts represented a major division in your populations that possibly pre-dates deglaciation, that would be good to mention here as justification for splitting up the populations into separate models.

Discussion

Lines 312-355: This section can be substantially reduced and moved to a less prominent place in the discussion. Distance to the sea is not a major factor that appears to be structuring these populations and it would be better to consider models that include the effects of the confounding factors you pointed out (i.e. effective population size) rather than present model results with “outlier” sites removed.

Lines 339-341: Estimates of effective population sizes would strengthen your inferences about the potential influence of genetic drift in your populations.

Line 363: You did not test for interaction effects between your environmental variables.

Lines 368-370: Are “hydro schemes” dams? I would use a more specific descriptive term here.

Lines 368-376: It seems obvious to me that dams would have very different effects on connectivity compared to irrigation canals and dykes. If you have this information, why not parameterize your models specifically to look at the effects of dams versus other waterbody modifications?

Lines 378-408: This section can be substantially reduced.

Line 410: Rename this section. As presented, it implies that the previous parts of the discussion only applied to the east coast.

**Do you want your identity to be public for this peer review?** For information about this choice, including consent withdrawal, please see our Privacy Policy

Reviewer #1: No

Reviewer #2: No

Reviewer #3: No

Reviewer #4: **Yes: ** Justin Waraniak

---

## [Author Response · Author response to Decision Letter 2]

22 May 2025

Dear Finn Cowell; The review of revised MS you submitted to PLOS ONE has completed. To reach the current decision, we invited additional reviewers to review the revised MS. Referees provided us with very valuable comments, I warmly appreciate for their agreement to review this submission and for very valuable comments. I would like to ask you revising your MS for another time according to the comments appended below. In addition to the referees’ comments, please consider those of mine in your revision:

1- I agree with R#4 to re-locate Table S1 in the main body of the paper.

2- Describe the methods in detail.

3- Follow the guide from PLOS ONE for the use of map: it needs to indicate copyright.

4- For figures 2 and 3: Increase the font size for numbers shown in X-axis and Y-axis as well as title for each panel.

Response:

We are grateful for the feedback received on this submission and we have altered our manuscript to comply with the above comments. We thank both yourself and the reviewers for their time and effort and have made a number of changes to the manuscript, as well as responding to all the reviewers’ comments.

Reviewer #1: I enjoyed reading the manuscript for a second time, with a bit more perspective. As stated in my original review, I was less familiar with some of the methods and approaches used here, so I apologize to the authors if my original review lacked nuance at times. I want to thank them for their enlightening, if not humbling, response.

I consider this study to offer a valuable contribution to the field, and now that my concerns have been addressed, I recommend its publication in its current form.

Response:

We take this opportunity to once again thank this reviewer for their initial comments, many of which helped us to greatly improve the manuscript. We are glad to hear that our responses were interesting.

Reviewer #2: I have enclosed more comments in two files. One is a response to your response to my review and the other provides some additional commentary on your manuscript (I used the section with your original manuscript and your responses to the reviews). My fundamental criticism was that your selected metrics don’t have the variability necessary to tease out the mechanisms of population structure. Your overall goal and the methods are not at issue. You have a great looking genetics data set and I am confident that environment metrics exist that can help delineate mechanisms of structuring.

Response:

We thank this reviewer for their continued engagement with our manuscript and attempt to address the issues raised by them. We respectfully disagree with this reviewer’s central claim, which is that the metrics we use lack sufficient variability. We go through the comments in response to our manuscript followed by their response to our initial response. The line numbers we have inserted as part of the comments that reviewer two made on the marked up version of the manuscript.

Lines 117-121: Clarity - It appears there are 7 groups and 48 sites. There are differences in the number of rivers/populations within each group (presumably) - tell us how you ensured that no group was over represented in the analyses, e.g., if there there had been 4 genetic groups (clusters of rivers/populations), then you sampled n=12 from each group and these 12 were randomly selected within a group (n = 48).

Response:

Some of the genetic clusters encompassed only a few sampling sites over a very limited geographic area. An equal number of sites per genetic cluster would therefore have resulted in a very small geographic area being overrepresented. We therefore had to sample from all genetic clusters, ensuring sites were not geographically concentrated, and the sites chosen are a reflection of our attempts to strike this balance.

Lines 194-195: The inclusion of elevation is important in a metric of river slope that reflects the probability of accessing the upper reaches, i.e., as important as distance, which needs to be included, assessed, and dismissed if that is the finding.

Response:

Given that the fundamental criticism from this reviewer is that our metrics are not variable enough, we now consider if the proposed alternative of river slope is in fact more variable than distance to the sea. The coefficient of variation for distance to the sea is equal to 0.9, while for river slope this only equals 0.486. The variance of the logarithm of distance to the sea is 1.91, while for river slope this same measure of variability is equal to 0.756. Lewontin (1966) notes a ratio of the latter metric can be compared to the F distribution. In this case the F value at 47 degrees of freedom, and using 0.05 as the significance threshold, is less than the ratio of 2.53 which we calculated. Therefore, not only are these measures of variability showing that distance to the sea is more variable than the river slope, our F-test suggests that it is significantly more variable. We do not therefore think it is worth rerunning a computationally taxing GESTE run to include a less variable metric, when low variability is the chief concern of this reviewer.

We will also take this opportunity to point out that we do not claim that this study has considered all possible metrics of interest. We do however consider a great many, and from our perspective, the statistical effect (small or large) that each of these have on genetic diversity and differentiation is of interest, even if other influences are yet to be understood.

Lines 306-308: Here is an example of the over-interpretation of results. There is no evidence in this study that waterbody modification had an "effect". You did see a correlation between these observations that suggest a relationship, but without a controlled experiment you can't know its an effect, and additionally, "non-negligible" is not the correct term, "apparent" is appropriate in all of our field studies.

Response:

We respectfully disagree that we are overinterpreting our results because we are not referring to a causal effect. It is common practice to use the word effect to describe a statistical effect. Michael Morrissey’s book Elementary Statistical Modelling has the following words to say on the subject:

‘ In every day usage, the wording “A has an effect on B” may well be taken to mean A has a causal effect on B. This very strong meaning is not typically intended when people use the word “effect”. This is just typical terminology typically used by biologists to describe different kinds of associations between variables (e.g., as can be expressed by the slope of a regression line). ‘

Line 323: The alternative interpretation as I first suggested is that your metrics don't have the variability necessary to expose mechanisms; the class variables are too coarse (as you recognize) and your numerical metrics vary too little. In both cases this impacts the mathematics (the strength of your predictability which depends on variation) and the biological relevance of the selected metrics, i.e., the animals' range of tolerances (their behaviour over time) likely exceed the environmental variation you are using in the analyses (the temperature comment in my first review).

Response:

This comment is made in response to our distance to the sea section of the discussion and we believe we have addressed the fact that the river slope is less variable than distance to the sea. We are sure that some of the class variables we have used might have information underlying them that would be of interest. The class variables were however obtained from databases created by researchers with expertise in environmental monitoring. As such they encapsulate all the variables that were considered as relevant for describing the extent to which the river has been affected by human activities. We therefore expect a waterbody with good water quality to be different from one categorised as bad. We also expect heavily modified water bodies to differ from those that are not, although we admit that the way in which water bodies are modified will vary.

Finally, this reviewer takes issue with our temperature metric as the temperatures do not exceed the animals’ range of tolerances. We used the maximum daily temperature on the hottest year for the last 20 years. This is a metric that the Scotland River Temperature Monitoring Network (SRTMN) made available and we do not know of any other source of nation-wide river temperature estimates for specific stretches of river that are reliable. The maximum temperature over a shorter time period than a day is not something made available by the SRTMN so we believe this is the best metric that can be used. Finally, we need to stress that the rationale for the choice of maximum temperature is that extreme events can cause events like major fish mortality or inaccessibility to sections of river, as hypothesised in Hand et al. (2016).

We mention in our discussion that previous studies have shown increased genetic differentiation in areas of high temperature, while others show reduced differentiation. A lack of variability is therefore not the only explanation for temperature not being found to greatly influence genetic differentiation. The causal mechanism might not be consistent. Consider the hypothesising done by Hand et al. (2016) in their population genetic study of anadromous Oncorhynchus mykiss. They considered the possibility that ‘ Higher stream temperatures lead directly to increased mortality or indirectly through increased physiological stress during return migration for adults. Thus, increased mortality leads to fewer returning breeders and lower local Ne ‘ while also acknowledging the alternative possibility that ‘ Higher stream temperatures lead to blocked routes of migration to natal streams further increasing straying rates leading to lower FST ‘. That is to say, the first biological mechanism of Hand et al. (2016) might be at work in some areas, while the other mechanism might be counterpoising this in other areas. Incidentally, Hand et al. (2016) used maximum daily temperatures as well.

Lines 324- 340: This is quite a bit of logic gymnastics to explain your results which as I have tried to point out, remain inconclusive with respect to your recolonization progression question.

Response:

This section of the discussion may well be too long and, since two reviewers drew our attention to this section, with reviewer 4 suggesting that we cut it down, we do so. We appreciate that there remain some unexplained results, but we do not believe this to be a major issue as this study is not all encompassing.

Line 363: correlation?

Response:

This section has now been removed from this discussion after considering feedback from both reviewers 2 and 4.

Line 379-383: Here is an example of the coarseness of scale affecting your results and interpretation. It is the efficiency of fish passage that will impact genetic structure as many studies demonstrate. That is the metric that would have given you the power of resolution you are seeking.

Response:

We consider this to be a surprisingly difficult variable to measure in a quantitative way. Firstly, hydro schemes do not simply act as a barrier that obstruct upstream passage. For example, they also abstract water which reduces the carrying capacity of the river below a dam and this is likely to have an impact on genetic diversity and differentiation. This is relevant for several of the sites we consider. Some dams are also more easily passed than others but quantifying this may be difficult.

Another consideration to factor in is historical stocking that occurred above dams in Scotland and continues to occur in the form of mitigation stocking. Poor records make it almost impossible to consider the impact of the former, while the latter is also difficult as brood stock are often caught afresh each year and this likely involves capturing individuals which originated from elsewhere in the catchment, increasing gene flow. This could counterpose the reduced gene flow from the barriers. We therefore think that, given current knowledge, any measure of passage efficiency would be unreliable and likely to introduce biases in our analyses.

Line 416: This is the reason to include a metric like river slope which was available to you.

Response:

We address this above where we show that river slope is less variable than distance to the sea.

Reviewer #2 response to our initial response.

The lack of variability isn’t ‘apparent’ it is obvious, and the authors make reference to it throughout.

Response:

We suggest that these are limitations that are well within the realms of acceptability. Often the reduced variability that we make reference to is a feature of the landscape that these fish occupy, such as the reduced distance from the sea on the west coast compared to the east coast. In other words, Scottish rivers and streams do not exhibit much variation in the environmental variables to which we had access.

As the authors point out, categorical variables by there very nature mask the natural variation in the parameters they represent.

Response:

Categorical variables mask some of the variation but certainly not all of it. We do not believe that this restriction in variability results in there being no utility to using these categorical variables.

I agree that these standard metrics are used and useful, but they mask the variation in the natural environment that are the mechanism of selection and population structure the authors seek to explain.

Response:

This is again a limitation that we understand, and which biologists for hundreds of years have understood, and this has not stopped us, or the biologists down the years, from utilising categorical variables. We do not understand why, in our case, it is wrong for us to utilise categorical variables.

To the authors – this type of passive aggressiveness doesn’t help your argument. In this case, I have not decided anything. I have pointed out that the temperature data (the metric you selected) varies too little to be a useful metric – mathematically and biologically. You conclude that your temperature metric provided no support for population structuring and you correctly point out that temperature is a critical aspect of structuring. My comment is unchanged: river temperature is important, but the metric selected doesn’t have the variability necessary to help discern underlying mechanisms of population structuring.

Response:

Firstly, we would like to apologise for any offence caused to this reviewer and can say with all honesty that no passive aggressive tone was intended in what was a clumsily worded sentence on our part. We do however respectfully disagree with this reviewer that our temperature metric is inappropriate and we have hopefully addressed this above to some extent. We will however add here that, although we write about the impact of temperature in other population genetic studies, it does not logically follow that we are going to find a similar result in different populations experiencing a different temperature regime.

Another note to the authors. You have incorrectly assumed that I don’t know Scottish rivers and their landscapes.

Response:

Apologies.

I used the distance a salmon can travel comment to point out that this metric is more nuanced than distance alone, e.g., river slope, numbers and location of natural/human barriers. You haven’t captured these elements of the riverscape by distance alone.

Response:

We have addressed some of these points above, but we would like to stress here that our study is not exhaustive and we chose variables based upon what we thought might have an effect on genetic diversity and differentiation, which in turn was based on previously published papers.

I wasn’t clear the first time – altitude gives you a river slope which is an important metric you could have included.

Response:

Now understood and hopefully addressed above.

I am not asking you to find variation where it doesn’t exist; I am pointing out that there are metric with better resolution that will give you the variability necessary to tease out the mechanisms of population structuring.

Response:

We are interest

---

## [Decision Letter · Decision Letter 2]

30 Jun 2025

Dear Dr. Cowell,

We look forward to receiving your revised manuscript.

Kind regards,

Sayyed Mohammad Hadi Alavi

Academic Editor

PLOS ONE

Journal Requirements:

Additional Editor Comments:

Dear Dr. Finn Cowell; Thank you very much for your revision. Please let me inform you that your submission needs a minor revision before being accepted for publication. In particular, (1) Introduction needs in-brief description on anthropogenic factors that affect genetic diversity and structuring of salmonid populations; (2) Dr Waraniak suggested to run a variance partitioning analysis. I would be very much appreciate it if you kindly revise the manuscript according to the comments appended below.

Very best regards

Hadi Alavi

Reviewers' comments:

Reviewer's Responses to Questions

**Comments to the Author**

Reviewer #4: (No Response)

2. Is the manuscript technically sound, and do the data support the conclusions?

Reviewer #4: Yes

3. Has the statistical analysis been performed appropriately and rigorously?

Reviewer #4: No

4. Have the authors made all data underlying the findings in their manuscript fully available?

Reviewer #4: Yes

5. Is the manuscript presented in an intelligible fashion and written in standard English?

Reviewer #4: Yes

Reviewer #4: The authors have improved the manuscript somewhat, though there are a few areas that I still have some concerns about before recommending this manuscript for publication. The main issue I have currently is that while the authors added a redundancy analysis as I recommended, they perhaps did not understand the purpose of the RDA and implemented it in a way that does not make much sense and does not provide many new insights compared to the other analyses they present. Additionally, I do not think the authors have fully addressed my concerns around specificity and providing enough appropriate details (particularly in the introduction) on the environmental factors that would allow the audience to make clear links between patterns in the genetic data and the environmental variables and assess how the analyses have been interpreted for themselves. More specific suggestions are listed below:

Major issues

Introduction (Lines 82-105): This listing of the anthropogenic factors is too generic and needs to be more explicit in how the authors might expect these factors to affect the genetic diversity and structuring of salmonid populations especially for the more general audience of PlosONE. E.g. in lines 96-99, dams and aquaculture escapees are introduced in the same sentence saying that they may impact genetic structuring, but the expected effects of these are very different, with dams increasing the risk of inbreeding (lower genetic diversity, possibly greater genetic differentiation) and aquaculture/stocking increasing the risk of outbreeding (increasing genetic diversity and possibly reducing genetic differentiation if the same stocking source interbreeds with different wild populations).

Methods (Lines 235-239): Using RDA in this way is not invalid per se, but is not substantially different from the GLMs. In genetic studies, it is more common to use a matrix of raw genotypes as the dependent variables in an RDA rather than a matrix of summary statistics. RDA is really meant to analyze high-dimensional data (e.g. genotypes from many markers), so using it for two summary statistics does not make a lot of sense. The authors may also wish to run a variance partitioning analysis (see Capblanq & Forester 2021: https://doi.org/10.1111/2041-210X.13722) as the most relevant way to present the results of the RDA for the purposes of their paper. This method will allow you to quantify how much inter-population genetic diversity is explained by your contemporary vs. historical factors and how much may be confounded by collinearity between your environmental factors. You will also be able to see if there are certain microsatellites that appear to be affected by specific environmental factors.

Minor suggestions

Line 57: replace “and” before Norway with a comma

Line 123-124: reference Figure 1 here

Lines 145-149: Figure legend should reference what the colors are depicting

Line 146: “Table S1” should now be “Table 1”

Line 215: Were there clear best models (∆AIC > 2 ) for each of the sets of GLMs tested? Also, considering that there are only 48 sites included, AICc for small sample sizes might be more appropriate in this case.

Lines 322-323: You can calculate the significance of individual explanatory variables in an RDA by running the “anova.cca” function with the ‘by = “terms”’ option.

**Do you want your identity to be public for this peer review?** For information about this choice, including consent withdrawal, please see our Privacy Policy

Reviewer #4: **Yes: ** Justin Waraniak

---

## [Author Response · Author response to Decision Letter 3]

15 Aug 2025

Dear Dr. Finn Cowell; Thank you very much for your revision. Please let me inform you that your submission needs a minor revision before being accepted for publication. In particular, (1) Introduction needs in-brief description on anthropogenic factors that affect genetic diversity and structuring of salmonid populations; (2) Dr Waraniak suggested to run a variance partitioning analysis. I would be very much appreciate it if you kindly revise the manuscript according to the comments appended below.

Response:

We are grateful for the feedback received on this submission and we have altered our manuscript to comply with the comments. Please, note that in the process of revising the ms, we realised that an input file for calculating allelic richness did not use the proper coding for missing data. We have corrected this mistake and can confirm that the new results did not change the main conclusions of our analysis.

Reviewer #4: The authors have improved the manuscript somewhat, though there are a few areas that I still have some concerns about before recommending this manuscript for publication. The main issue I have currently is that while the authors added a redundancy analysis as I recommended, they perhaps did not understand the purpose of the RDA and implemented it in a way that does not make much sense and does not provide many new insights compared to the other analyses they present. Additionally, I do not think the authors have fully addressed my concerns around specificity and providing enough appropriate details (particularly in the introduction) on the environmental factors that would allow the audience to make clear links between patterns in the genetic data and the environmental variables and assess how the analyses have been interpreted for themselves. More specific suggestions are listed below:

Response:

We thank this reviewer for their continued engagement with our manuscript and the useful comments. We hope the changes we have made in accordance with specific suggestions will allay any concerns.

Reviewer #4:

Major issues

Introduction (Lines 82-105): This listing of the anthropogenic factors is too generic and needs to be more explicit in how the authors might expect these factors to affect the genetic diversity and structuring of salmonid populations especially for the more general audience of PlosONE. E.g. in lines 96-99, dams and aquaculture escapees are introduced in the same sentence saying that they may impact genetic structuring, but the expected effects of these are very different, with dams increasing the risk of inbreeding (lower genetic diversity, possibly greater genetic differentiation) and aquaculture/stocking increasing the risk of outbreeding (increasing genetic diversity and possibly reducing genetic differentiation if the same stocking source interbreeds with different wild populations).

Response:

We have linked each of the anthropogenic factors to the evolutionary forces that we expect to have an impact on genetic differentiation and diversity (lines 96-101 and 107-109).

Reviewer #4:

Methods (Lines 235-239): Using RDA in this way is not invalid per se, but is not substantially different from the GLMs. In genetic studies, it is more common to use a matrix of raw genotypes as the dependent variables in an RDA rather than a matrix of summary statistics. RDA is really meant to analyze high-dimensional data (e.g. genotypes from many markers), so using it for two summary statistics does not make a lot of sense. The authors may also wish to run a variance partitioning analysis (see Capblanq & Forester 2021: https://doi.org/10.1111/2041-210X.13722) as the most relevant way to present the results of the RDA for the purposes of their paper. This method will allow you to quantify how much inter-population genetic diversity is explained by your contemporary vs. historical factors and how much may be confounded by collinearity between your environmental factors. You will also be able to see if there are certain microsatellites that appear to be affected by specific environmental factors.

Response:

We agree with the reviewer that RDA is typically applied to the genotype matrix, but this is only possible when using biallelic SNPs, in which case genotypes can be coded unequivocally as (0, 1, 2) depending on how many copies of the reference alleles an individual has at a given locus. However, we are working with microsatellites with several alleles at each locus and, therefore we cannot use this simple coding to assign values to each genotype. There is no ideal solution for coding microsatellites in a way that makes them amenable to use in an RDA. Thus, we used an approach implemented in hierfstat, which consists in generating a total of 732 dependent variables by converting the microsatellite data into allele frequencies, with one allele (column) removed for each locus. Three RDA models were run: the first had all variables, the second included contemporary variables and controlled for the effect of the historical variables and the third included historical variables and controlled for the contemporary variables. In this instance, distance to the sea and time since deglaciation were the historical variables. As can be seen in table 1, none of these have a statistically significant impact on the differing frequencies of alleles between populations. We do not think these results are particularly interesting and therefore we have decided not to include them in the manuscript.

Table.1: Results of a variance partitioning analysis using RDA.

Variables Canonical Variance R2 p-value Proportion of Canonical Variance

All 140 0.191 0.504 1

Contemporary 96.7 0.132 0.223 0.691

Historical 31.6 0.043 0.422 0.226

Reviewer #4:

Minor suggestions

Line 57: replace “and” before Norway with a comma

Response:

Done.

Reviewer #4:

Line 123-124: reference Figure 1 here

Response:

Done.

Reviewer #4:

Lines 145-149: Figure legend should reference what the colors are depicting

Response:

Done.

Reviewer #4:

Line 146: “Table S1” should now be “Table 1”

Response:

Done.

Reviewer #4:

Line 215: Were there clear best models (∆AIC > 2 ) for each of the sets of GLMs tested? Also, considering that there are only 48 sites included, AICc for small sample sizes might be more appropriate in this case.

Response:

The range of AIC values is > 2 but the difference between the lowest AIC value and the next lowest is < 2. We switched to using AICc, as suggested, and the result of this is that the simplest model was chosen in each case. For all sites and the west coast this resulted in the time since deglaciation being left in the model. For the east coast, waterbody modification was chosen (see table 2 and figure 2).

Reviewer #4:

Lines 322-323: You can calculate the significance of individual explanatory variables in an RDA by running the “anova.cca” function with the ‘by = “terms”’ option.

Response:

We thank the reviewer for pointing this out as an option open to us, but we are happy with only presenting the significance between matrices Y and X and then presenting the loadings of the variables for each canonical axis, with the significance of the axes shown in figure 4. It can clearly be seen in table 5 that waterbody modification and time since deglaciation have far greater loadings than any of the other variables. Therefore, we think that all the appropriate information is already in the paper and no more detail is required. We would also like to point out that only time since deglaciation and waterbody modification have significant p-values so the results we are presenting are in no way misleading.

---

## [Decision Letter · Decision Letter 3]

4 Sep 2025

Dear Dr. Cowell,

Thank you for submitting your manuscript to PLOS ONE. After careful consideration, we feel that it has merit but does not fully meet PLOS ONE’s publication criteria as it currently stands. Therefore, we invite you to submit a revised version of the manuscript that addresses the points raised during the review process.

We look forward to receiving your revised manuscript.

Kind regards,

Sayyed Mohammad Hadi Alavi

Academic Editor

PLOS ONE

Journal Requirements:

**Additional Editor Comments:**

Dear Finn Cowell:

I would appreciate if you kindly perform the last revision on your submission. The reviewer asked for some minor revision.

Very best regards

Hadi

Reviewers' comments:

Reviewer's Responses to Questions

**Comments to the Author**

Reviewer #4: (No Response)

2. Is the manuscript technically sound, and do the data support the conclusions?

Reviewer #4: Yes

3. Has the statistical analysis been performed appropriately and rigorously?

Reviewer #4: Yes

4. Have the authors made all data underlying the findings in their manuscript fully available?

Reviewer #4: Yes

5. Is the manuscript presented in an intelligible fashion and written in standard English?

Reviewer #4: Yes

Reviewer #4: The authors have addressed most of my concerns from the previous round of reviews. I only have two small comments:

1) While it is possible to use multi-allelic loci in an RDA with each allele coded as 0/1/2 (so long as the organism is a diploid; this refers to the count of a specific allele in a genotype, regardless of whether a loci only has 2 possible alleles like a biallelic SNP or more like a multi-allelic microsat), conducting a population-wise RDA using allele frequencies is an equally valid approach that would lead to similar conclusions since the environmental variables were measured at the population level. While the authors may choose not to report the results of that analysis, as there were no significant relationships found between allele frequency patterns and the environmental variables tested, I would consider that result somewhat interesting as a suggestion that isolation by environment has not been a strong force shaping the population structure of these populations. Additionally, the RDA on allelic richness and Fst that was included does not make much sense in this context and I would recommend removing it from the manuscript.

2) If you are going to use AIC (or AICc? It was unclear whether the model selection procedure methods were updated) as a way to assess models, those results need to be reported at some point in the manuscript. If there was one clear best-performing model in each case, you can minimally report that ∆AIC >2 for all other models tested.

**Do you want your identity to be public for this peer review?** For information about this choice, including consent withdrawal, please see our Privacy Policy

Reviewer #4: No

---

## [Author Response · Author response to Decision Letter 4]

9 Sep 2025

Reviewer #4: The authors have addressed most of my concerns from the previous round of reviews. I only have two small comments:

1) While it is possible to use multi-allelic loci in an RDA with each allele coded as 0/1/2 (so long as the organism is a diploid; this refers to the count of a specific allele in a genotype, regardless of whether a loci only has 2 possible alleles like a biallelic SNP or more like a multi-allelic microsat), conducting a population-wise RDA using allele frequencies is an equally valid approach that would lead to similar conclusions since the environmental variables were measured at the population level. While the authors may choose not to report the results of that analysis, as there were no significant relationships found between allele frequency patterns and the environmental variables tested, I would consider that result somewhat interesting as a suggestion that isolation by environment has not been a strong force shaping the population structure of these populations. Additionally, the RDA on allelic richness and Fst that was included does not make much sense in this context and I would recommend removing it from the manuscript.

response:

We thank the reviewer for giving us the option on whether or not to include the results of this analysis and we note the point this reviewer makes. However, fundamentally the results from the allele frequency RDA does not change the conclusions that can be drawn from the other analyses already included in the manuscript so we do not include it in this latest version. We do however take the advice of this reviewer and remove the Ar and Fst RDA from the manuscript.

2) If you are going to use AIC (or AICc? It was unclear whether the model selection procedure methods were updated) as a way to assess models, those results need to be reported at some point in the manuscript. If there was one clear best-performing model in each case, you can minimally report that ∆AIC >2 for all other models tested.

response:

We take this opportunity to clarify that we are using AICc scores (line 220). We have now also included a sentence stating that “The difference between the AICc score of the simplest model and the next lowest score was not > 2 but the simplest model was nonetheless chosen as the best model in each case” (lines 265-266). We hope that this is alright because we would rather not have an entire table dedicated to AICc scores for each model, including model specifications, as this would be very large (5 x 26) and most of the information would be surplus to requirements.

---

## [Editor Report · Decision Letter 4]

11 Sep 2025

Post-Pleistocene Colonisation Rather than the Contemporary Environment has Most Influenced the Current Population Structure of Scottish Atlantic Salmon (Salmo salar)

PONE-D-24-37086R4

Dear Dr. Cowell,

We’re pleased to inform you that your manuscript has been judged scientifically suitable for publication and will be formally accepted for publication once it meets all outstanding technical requirements.

Kind regards,

Sayyed Mohammad Hadi Alavi

Academic Editor

PLOS ONE

Additional Editor Comments (optional):

Authors revised the MS properly.
---

## [Editor Report · Acceptance letter]

PONE-D-24-37086R4

PLOS ONE

Dear Dr. Cowell,

I'm pleased to inform you that your manuscript has been deemed suitable for publication in PLOS ONE. Congratulations! Your manuscript is now being handed over to our production team.

Kind regards,

on behalf of

Dr. Sayyed Mohammad Hadi Alavi

Academic Editor

PLOS ONE